# Iron phthalocyanine on Au(111) is a "non-Landau" Fermi liquid

R. Žitko [1,2 ✉], G. G. Blesio [1,3], L. O. Manuel [3] & A. A. Aligia[4]

The paradigm of Landau's Fermi liquid theory has been challenged with the finding of a strongly interacting Fermi liquid that cannot be adiabatically connected to a non-interacting system. A spin-1 two-channel Kondo impurity with anisotropy $D$ has a quantum phase transition between two topologically different Fermi liquids with a peak (dip) in the Fermi level for $D < D_c$ ($D > D_c$). Extending this theory to general multi-orbital problems with finite magnetic field, we reinterpret in a unified and consistent fashion several experimental studies of iron phthalocyanine molecules on Au(111) that were previously described in disconnected and conflicting ways. The differential conductance shows a zero-bias dip that widens when the molecule is lifted from the surface (reducing the Kondo couplings) and is transformed continuously into a peak under an applied magnetic field. We reproduce all features and propose an experiment to induce the topological transition.

[1] Jožef Stefan Institute, Jamova 39, SI-1000 Ljubljana, Slovenia. [2] Faculty of Mathematics and Physics, University of Ljubljana, Jadranska 19, SI-1000 Ljubljana, Slovenia. [3] Instituto de Física Rosario (CONICET) and Universidad Nacional de Rosario, Bv. 27 de Febrero 210 bis, 2000 Rosario, Argentina. [4] Instituto de Nanociencia y Nanotecnología CNEA-CONICET, Centro Atómico Bariloche and Instituto Balseiro, 8400 Bariloche, Argentina. ✉email: rok.zitko@ijs.si

Transport properties of single molecules in contact with metal electrodes are being extensively studied due to their potential use as active components of new electronic devices[1–3]. The paradigmatic Kondo effect (screening of molecule's magnetic moment by conduction band electrons from metal) is ubiquitous in these systems and allows to control the current by different external parameters[4–19].

The Kondo model (KM) for a molecule with spin $S$ coupled to $n$ conduction bands with different symmetries (channels) can be classified into three types: underscreened KM when the number of channels is too small to fully screen the spin, overscreened KM the other way, and compensated Kondo model (CKM) at the frontier. If there is only one way to build the spin, the frontier is at $n = 2S$[20]. The ground state of an underscreened molecule has a non-zero residual spin, and is a singular Fermi liquid with logarithmic corrections at low energies[21,22]. In particular, the case with $S = n = 1$ and residual spin 1/2 has been thoroughly investigated in molecular nanoscopic systems[8,10,13,22,23]. The overscreened model is a non-Fermi liquid that exhibits even more striking singular properties at low temperatures[24–26]. In general it was, however, strongly believed that the CKM always corresponds to an ordinary Fermi liquid (OFL) with regular low-energy behavior showing no non-analytical features. Nevertheless, it has been recently shown that this changes in the presence of an anisotropy term $DS_z^2$ (refs. [27,28]), which can drastically modify the ground state[29–33].

In a Fermi liquid, the life-time of the quasiparticles with excitation energy $\omega$ scales as $\omega^{-2}$ close to the Fermi level at zero temperature, due to restrictions of phase space imposed by the Pauli principle, as first shown by L. Landau. This picture applies to all weakly interacting metals, but it may be broken in the presence of strong interactions. In particular, in the overscreened KM the quasiparticles are not well-defined even at the Fermi level, and in the underscreened KM they have a logarithmic dependence on $\omega$ for small $\omega$[21,22]. The CKM with $D = 0$ (and the more general Anderson impurity model from which the CKM is derived) corresponds to an OFL. In this case, the Friedel sum rule relates the zero-temperature spectral density of the localized states at the Fermi level ($\omega = 0$), as well as the zero-bias conductance, with the ground-state occupancy of these states[34,35]. As a consequence of the fractional occupancies per channel and spin in the Kondo regime, the spectral density has a Kondo peak at the Fermi level and the differential conductance d$I$/d$V$ a zero-bias anomaly, i.e., a peak at voltage $V = 0$.

However, for the $S = 1$ CKM, this picture changes dramatically for $D > D_c$, where $D_c$ is the critical anisotropy at which a topological quantum phase transition takes place. The spectral density at $\omega = 0$ vanishes. In this case, the Friedel sum rule should be modified to allow for a non-zero value of a Luttinger integral $I_L$, which has a topological character with a discrete set of possible values. A system in this regime is a Fermi liquid, yet it cannot be adiabatically connected to a non-interacting system. A closer scrutiny reveals the presence of a $\delta$-peak exactly at the Fermi level in the imaginary part of the impurity self-energy, even though the low-energy scattering properties are not in any way anomalous and remain Fermi-liquid like. Such a ground state has been called a non-Landau Fermi liquid (NLFL)[27,28]. Previously, other strongly interacting models have been shown to display NLFL behavior for some parameters[36,37].

A set of recent spectroscopic measurements performed on iron phthalocyanine (FePc) on Au(111) showed features reminiscent of those expected in NLFL systems. Two relevant experiments, performed before the theory for the NLFL system was developed[15,18], reported basic characteristics of the system. Another more recent work[19] is a particularly revealing detailed study of the magnetic field and temperature dependence. The existing interpretations proposed in these three works have serious deficiencies, as we discuss in the following. The NLFL theory accounts for the totality of the available results on FePc/Au(111), as well as for very recent experiments on MnPc/Au(111)[38]. A non-trivial extension of the existing theory to nonequivalent channels and finite magnetic fields presented in this work, was necessary to make these inferences. We also explain how to observe experimentally the topological quantum phase transition between a non-Landau and a Landau Fermi liquid.

Ab initio calculations[15] show that the molecule in the "on top" position has a spin $S = 1$ formed by an electron in the 3d orbital of Fe with symmetry $3z^2 - r^2$ and another shared between the degenerate $\pi$ ($xz$, $yz$) Fe 3d orbitals. The former has larger hybridization with the Au substrate than the latter. The molecule can be described by a three-channel multi-orbital Anderson Hamiltonian, which is compensated[39]. The low-temperature differential conductance d$I$/d$V$ measured by a scanning-tunneling microscope (STM) at zero magnetic field shows a broad peak of half-width ~240 K[15,39] centered near $V = 0$ and superposed to it a narrow dip of half width ~2.7 K[18], also centered near $V = 0$. This is exactly the shape expected in the NLFL regime of the anisotropic $S = 1$ CKM near the topological quantum phase transition at $D = D_c$[27,28].

In the absence of these theoretical results, the first set of experiments[15] was initially interpreted as a two-stage Kondo effect, with the larger energy scale corresponding to the stronger hybridized channel with $3z^2 - r^2$ symmetry, and the smaller one due to the $\pi$ orbitals[15,39]. The magnetic anisotropy was not taken into account. In general, as a consequence of interference effects, a d$I$/d$V$ curve has a peak when the tip of the STM has a stronger hopping to the localized orbitals and a dip when the tip has instead a stronger hopping to the conduction electrons[40,41]. Therefore, this interpretation requires that the tip has a stronger (weaker) hybridization to the localized $3z^2 - r^2$ ($\pi$) orbitals than to the conduction electrons of the same symmetry in addition to appropriate magnitudes of these hybridizations[39]. This seems unlikely.

The more recent experiment in which the molecule is raised from the surface, thereby reducing the hybridization of both localized states to the Au substrate, provides a stringent test of the above interpretation[18]. In the original interpretation based on the two-stage KM, both the peak and the dip should narrow and become steeper because both Kondo temperatures decrease. Instead, the experiments show clearly that the dip broadens and the peak flattens as the molecule is raised, in agreement with the expectations for the NLFL (see below). The theoretical explanation of the authors, further discussed in Note 6 of the supplemental material of their paper, is only qualitative and considers two contributions with many parameters.

The study of magnetic field effects[19] unveils a surprising transformation of the narrow dip into a peak with increasing field strength $B$. This is a challenge to all existing theories so far. Based on ab initio calculations this effect has been interpreted as a rearrangement of the presence of localized orbitals of $3z^2 - r^2$ and $\pi$ symmetry near the Fermi level[19]. However, the difference between the energies of these states, of the order of 1 eV[15], is much larger than the Zeeman energy of about 10T. Furthermore, many-body effects, which play a crucial role by shifting the relative positions of peaks in the density of states[42] and in the Kondo effect[43] have been altogether neglected. Therefore this explanation is very unlikely.

A similar effect of $B$ has been observed in MnPc on Au(111) which has a similar occupancy of the 3d states as FePc. The authors of that work proposed an interpretation in terms of a quantum phase transition involving localized singlet states[38]. This transition, first discussed in the context of Tm impurities[44] and

more recently in quantum dots[45,46], in particular with $C_{60}$ molecules[9,13,47,48], might be qualitatively consistent with the experiment (see Fig. 10 of ref. [48]). However, this requires that the singlet be below the triplet by a few meV, while in fact the triplet is energetically favored by a Hund's coupling of the order of 1 eV[15].

In this work, we show that the behavior of FePc on Au(111) can be interpreted in a unified and consistent way in terms of an anisotropic $S = 1$ CKM in the parameter regime where the system is a NLFL. Using the numerical renormalization group (NRG) we solve the anisotropic $S = 1$ CKM for inequivalent channels in the presence of magnetic field and show that all phenomena observed in the above-mentioned experiments can be explained in a consistent and simple way. We underpin these calculations by generalizing the topological theory of the Friedel sum rule to the case in which channel and spin symmetries are broken. In particular, we show how the discontinuous transition at zero magnetic field governs the molecule's excitation spectrum at finite fields, and why the evolution of the spectral line shape from a dip to a peak with increasing field is nevertheless continuous.

## Results

### Topological quantum phase transition and generalized Friedel sum rule.
The essence of the atomic multi-orbital three-channel Anderson impurity model for FePc, which is difficult to handle numerically, is captured by the $S = 1$ two-channel KM with the Hamiltonian

$$H_K = \sum_{k\tau\sigma} \varepsilon_{k\tau} c_{k\tau\sigma}^\dagger c_{k\tau\sigma} + \sum_{k\tau\sigma\sigma'} \frac{J_\tau}{2} c_{k\tau\sigma}^\dagger (\vec{\sigma})_{\sigma\sigma'} c_{k\tau'\sigma'} \cdot \vec{S} \\ + DS_z^2 - BS_z, \qquad (1)$$

where $c_{k\tau\sigma}^\dagger$ creates an electron in the Au substrate with wave vector $k$, pseudospin $\tau$ (representing a channel with symmetry $3z^2 - r^2$ for $\tau = 1$ and $\pi$ for $\tau = -1$) and spin $\sigma$. The first term describes the substrate bands, the second the Kondo exchange with the localized spin $\vec{S}$ with exchange couplings $J_1 > J_{-1}$, the third term is the single-ion uniaxial magnetic anisotropy, and the last term is the effect of an applied magnetic field $B$. This model is equivalent to that used by Hiraoka et al.[18] for large Hund's coupling $J_H$. As we show below, this model with three parameters explains all the experimental findings in FePc (While three channels participate in FePc[39], including all of them in an NRG calculation would lead to a too fast increase in the Hilbert space as a function of the iteration rendering it impossible to obtain precise results). Its fundamental ingredients are well justified by ab initio and/or ligand field multiplet calculations: the existence of an almost integer electronic occupancy of the Fe ion[18,49], the spin $S = 1$ of the FePc molecule[15,18,19,50], the markedly different hybridizations of the $d_{z^2}$ and $d_\pi$ orbitals with the Au substrate[15,18,51], and the presence of a non-negligible easy-plane magnetic anisotropy. We also consider that the gold conduction bands have a constant density of states, as first-principle calculations[52] do not find any signature of sharp structures (with widths of the order of $T_K^{(1)}$ or less), that would modify sensibly the low energy Kondo physics, and we take the half band width $W = 1$ as the unit of energy.

It is worth to stress that, in this work, we are looking for a minimal model that, on one hand, allows to understand, in a unified and rather simple way, the low-energy behavior of FePc on Au(111), and that, on the other hand, is suitable for a numerically exact resolution at very low energies. More realistic and quantitative modeling for this and other transition metal phthalocyanines on metallic surfaces should take into account the full complexity of the 3d orbital manifold and the hybridization with the surface[53–57].

The generalized Friedel sum rule is more conveniently discussed in terms of an auxiliary two-channel Anderson model $H_A$ from which $H_K$ can be derived in the limit of total local occupancy pinned to two (one electron in each orbital). This model is

$$H_A = \sum_{\tau\sigma} \epsilon_\tau d_{\tau\sigma}^\dagger d_{\tau\sigma} + \sum_\tau U_\tau n_{\tau\uparrow} n_{\tau\downarrow} + \\ - J_H \vec{S}_1 \cdot \vec{S}_{-1} + DS_z^2 - BS_z \qquad (2) \\ + \sum_{k\tau\sigma} \varepsilon_{k\tau} c_{k\tau\sigma}^\dagger c_{k\tau\sigma} + \sum_{k\tau\sigma} \left( V_\tau c_{k\tau\sigma}^\dagger d_{\tau\sigma} + \text{H.c.} \right),$$

where $d_{\tau\sigma}^\dagger$ creates a hole with energy $\epsilon_\tau$ in the $d$ orbital $\tau$, $n_{\tau\sigma} = d_{\tau\sigma}^\dagger d_{\tau\sigma}$, and $n_\tau = \sum_\sigma n_{\tau\sigma}$. $\epsilon_\tau$ and $U_\tau$ are the energy level and the Coulomb repulsion chosen such that $\epsilon_\tau = -U_\tau/2$ to fix the occupancy in each orbital to one, while the hopping $V_\tau$ characterizes the tunneling between the localized and conduction states with symmetry $\tau$. The Hund's coupling $J_H$ is responsible for the formation of the $S = 1$ degree of freedom. The actual Coulomb interaction contains more terms[27,28,57], but they are irrelevant for realistic parameters for FePc. The two models are related by the Schrieffer-Wolff transformation, such that $J_\tau \propto V_\tau^2/U_\tau$ (further details are given in Supplementary Note 1).

The impurity spectral function per orbital and spin, at the Fermi level and $T = 0$, is related to the quasiparticle scattering phase shift $\delta_{\tau\sigma}$ by[34,58–62]

$$\rho_{\tau\sigma}(\omega = 0) = -\frac{1}{\pi} \text{Im} G_{\tau\sigma}^d(0) = \frac{1}{\pi\Delta_\tau} \sin^2 \delta_{\tau\sigma}, \qquad (3)$$

where $G_{\tau\sigma}^d(\omega)$ is the impurity Green's function $\langle\langle d_{\tau\sigma}; d_{\tau\sigma}^\dagger \rangle\rangle$ and $\Delta_\tau = \pi\sum_k |V_\tau|^2 \delta(\omega - \epsilon_k)$ is the hybridisation strength, assumed independent of energy. According to the generalized Friedel sum rule, for wide constant unperturbed conduction bands one has

$$\delta_{\tau\sigma} = \pi\langle n_{\tau\sigma}\rangle + I_{\tau\sigma}, I_{\tau\sigma} = \text{Im} \int_{-\infty}^0 d\omega \, G_{\tau\sigma}^d(\omega) \frac{\partial \Sigma_{\tau\sigma}^d(\omega)}{\partial \omega}, \qquad (4)$$

where $\Sigma_{\tau\sigma}^d(\omega)$ is the impurity self energy. Further details are given in Supplementary Note 2.

Until recently, based on the seminal perturbative calculation of Luttinger[60], it was generally assumed that the Luttinger integrals $I_{\tau\sigma}$ vanish (at least for $B = 0$). However, several cases are now known where $I_{\tau\sigma} = \pm\pi/2$[22,27,28,36,37], yet the quasiparticle scattering phase shifts show no low-energy singularities. In particular, in our case for $B = 0$ and both orbitals equivalent, the four $I_{\tau\sigma} = I_L$ are equal by symmetry and $I_L$ has a topological character, being equal to 0 for $D < D_c$ and $\pi/2$ for $D > D_c$[27,28]. In this case, Eqs. (3) and (4) imply that in the Kondo limit, where $\langle n_{\tau\sigma}\rangle = 1/2$, the spectral densities $\rho_{\tau\sigma}(0)$ jump from $\rho_0 = 1/(\pi\Delta_\tau)$ for $D < D_c$ to 0 for $D > D_c$. In the general case, when the channels are not equivalent and a magnetic field is present, we show in Supplementary Note 2 that the conservation laws imply that three topological quantities $T, T_\tau, T_\sigma$ can still be defined:

$$T = \sum_{\tau\sigma} I_{\tau\sigma}, \; T_\tau = \sum_{\tau\sigma} \tau I_{\tau\sigma}, \; T_\sigma = \sum_{\tau\sigma} \sigma I_{\tau\sigma}, \qquad (5)$$

where $\sigma = 1 \, (-1)$ for spin $\uparrow (\downarrow)$. Although previous perturbative calculations assumed $T = T_\tau = T_\sigma \equiv 0$[35], we find numerically that $T_\tau = T_\sigma \equiv 0$, but $T = 4I_0$ with $I_0$ equal to either 0 or $\pi/2$. This allows us to write the individual Luttinger integrals in the general form

$$I_{1\uparrow} = I_{-1\downarrow} = I_0 - \alpha, I_{1\downarrow} = I_{-1\uparrow} = I_0 + \alpha, \qquad (6)$$

where $\alpha(D, B)$ is unknown. For $B = 0$, by symmetry $I_{\tau\uparrow} = I_{\tau\downarrow}$ implying $\alpha = 0$, and therefore not only $T, T_\tau$ and $T_\sigma$ but also

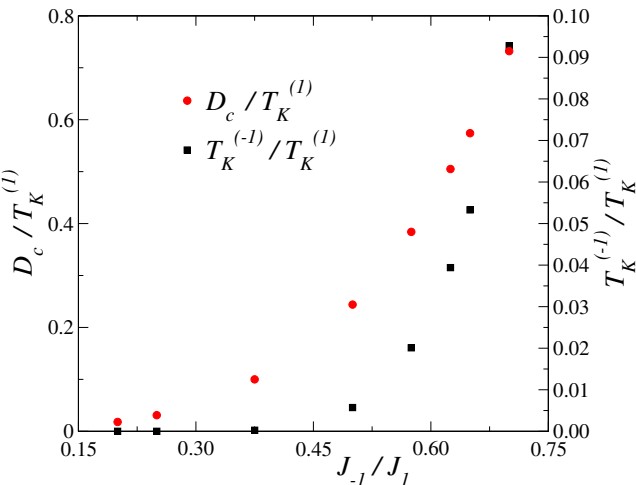

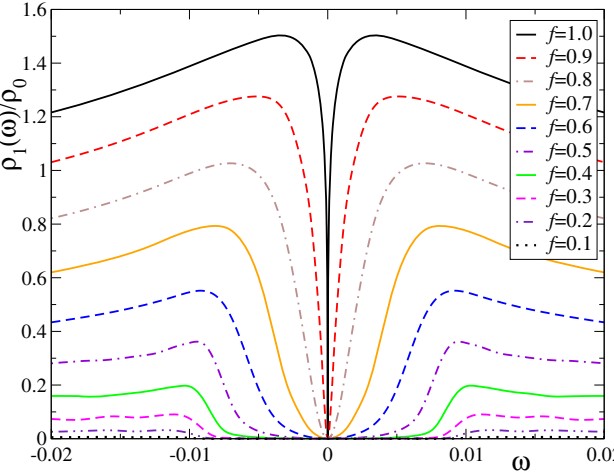

**Fig. 1 Characteristic energy scales.** Ratio of the lower Kondo temperature $T_K^{(-1)}$ over the higher Kondo temperature $T_K^{(1)}$ (right scale, red symbols) and ratio of the critical anisotropy $D_c$ over higher Kondo temperature (left scale, black squares), as a function of the smaller exchange coupling $J_{-1}$ at constant larger coupling $J_1 = 0.4$.

**Fig. 2 Variation of the spectral density as the molecule is raised from the surface.** The gradual decoupling from the surface is modeled through equal reduction of both exchange coupling constants by a factor $f$. The model parameters are $J_1 = 0.44 \times f$, $J_{-1} = 0.22 \times f$ and $D = 0.01$. See Supplementary Fig. 3 for a direct comparison with experiments.

individual Luttinger integrals $I_{\tau\sigma}$ have a topological character. The establishment of the general form in Eq. (6) and its numerical validation in the full parameter space of the model is one of the central results of this work.

We define the Kondo temperature $T_K^{(\tau)}$ as the temperature for which the contribution of channel $\tau$ to the zero-bias conductance for $D = B = 0$ falls to half of its zero-temperature value[27]. For the KM with channel symmetry [$J_{-1} = J_1$ in Eq. (1)], the topological transition takes place for $D_c \sim 2.5 T_K^{(1)}$[27], but $D_c$ decreases with decreasing $J_{-1}/J_1$ ratio. In fact, for $J_{-1} = 0$ it is known to be zero[23]. In FePc experimentally $D = 5$ meV[18] and $T_K^{(1)} \sim 20$ meV[15]. In Fig. 1 we represent the ratios $D_c/T_K^{(1)}$ and $T_K^{(-1)}/T_K^{(1)}$ for intermediate values of $J_{-1}/J_1$. The higher Kondo temperature $T_K^{(1)}$ changes only moderately as $J_{-1}/J_1$ varies, from 0.024 for $J_{-1}/J_1 = 0.2$ to 0.018 for $J_{-1}/J_1 = 0.7$ as a consequence of the competition between both channels[39]. Instead, $D_c$ and particularly $T_K^{(-1)}$ have a strong dependence on $J_{-1}$ in the range considered.

For comparison with the experiment, we take $D/W = 0.01$ (that means $W \sim 500$ meV[52], a different choice of $W$ does not affect the results in a sensitive manner) and fix $J_1 = 0.44$ so that $T_K^{(1)} \sim 200$ K as in the experiment. The remaining parameter was fixed at $J_{-1} = 0.22$ so that the system is in the non-Landau phase ($D > D_c$), with the width of the dip near to the experimentally observed one (as described later). The resulting Kondo temperatures are $T_K^{(1)} = 198$ K and $T_K^{(-1)} = 1.77$ K, while $D_c = 0.00950 \sim 4.7$ meV $\sim 55$ K.

We choose as the corresponding parameters in the auxiliary Anderson model $J_H = 0.1$, $U_\tau = 0.4$, $\epsilon_\tau = -U_\tau/2$, $\Delta_1 = 0.06$, $\Delta_{-1} = 0.0345$. The two models have the same low-energy behavior if the results are rescaled in terms of $T_K^{(1)}$ or $D_c$.

### Differential conductance at zero temperature in the absence of magnetic field.
The $s$-like orbitals of the STM tip have a larger overlap with the $3z^2 - r^2$ orbital of FePc than with the molecular $\pi$ orbitals and the substrate conduction-band wave functions. Ab initio calculations for Co on Cu(111) confirm this picture[63]. For this reason, the dominant contribution to the experimental d$I$/d$V$ spectra corresponds to the spectral density $\rho_1(\omega) = \rho_{1\uparrow}(\omega) +$

$\rho_{1\downarrow}(\omega)$ in channel $\tau = 1$. This description is both simpler and more physically realistic than the assumptions underlying the alternative interpretations of the measured d$I$/d$V$ from refs. [15,18,19]. The observed weak asymmetry in the line shapes indicates that some interference effects involving conduction orbitals are present. They will be incorporated in the next subsections. Here they do not modify the essential features and the conduction orbitals become less important as the molecule is raised from the surface.

Our result for FePc in the relaxed geometry corresponds to the black full line of Fig. 2 and reproduces the main features of the observed differential conductance except for some asymmetry in the experiments, which we neglect in this subsection as explained above. As the molecule is raised from the substrate by the attractive force of the STM tip, the Kondo coupling to the substrate is reduced[18]. We assume that both exchange coupling constants $J_\tau$ are reduced by the same factor $f$ because the same power-law dependence of the tunneling parameters $V_\tau$ with the distance is expected between the localized 3d orbitals and the extended conduction states[64]. As this factor $f$ decreases from 1, the spectral density flattens and the dip broadens so that the line-shape becomes similar to that observed in single-channel Kondo systems with magnetic anisotropy[30,31]. The exact same trend is observed experimentally[18]. For very small $f$, there should be two sharp steps at the unrenormalized threshold for inelastic spin-flip excitations ($\omega = \pm D$); in our calculations they are overbroadened due to technical reasons that limit the resolution of the NRG at large energies[65]. A detailed comparison with experiment is contained in Supplementary Note 4.

It would be interesting to push the STM tip against the molecule and tune from the tunneling to the contact regime. In this case the factor $f$ is expected to increase beyond 1[66–68], driving the system through the topological quantum phase transition to the OFL regime. This would be signaled by the sudden transformation of the dip into a peak of similar amplitude, i.e., compared to the baseline density of states, the height of the peak just after the transition should be similar as the depth of the dip just before the transition. This is an important prediction of our theory that should be the target of future experiments (although

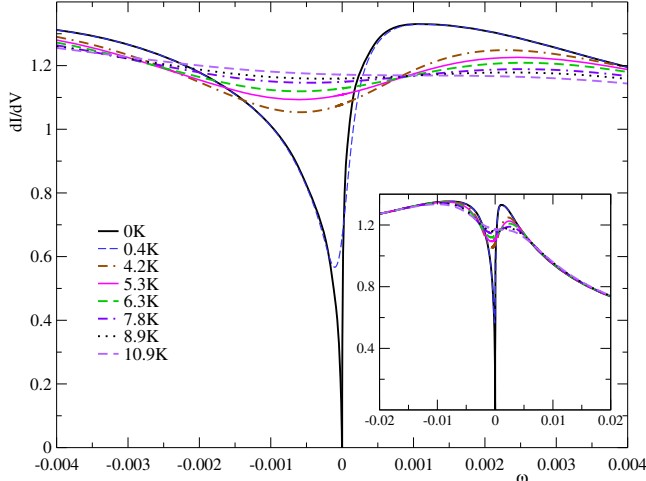

**Fig. 3 Temperature dependence of the differential conductance.** The main panel shows the close-up on the low-energy regions, while the inset shows an extended range. Parameters as in Fig. 2. See Supplementary Fig. 4 for a direct comparison with experiments.

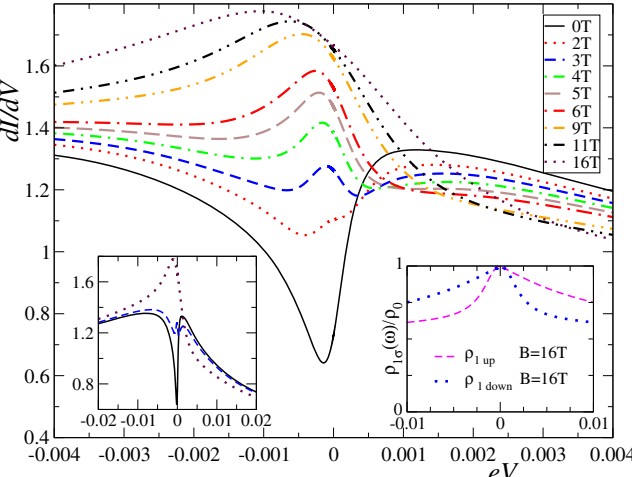

**Fig. 4 Magnetic field dependence of the differential conductance.** Left inset: differential conductance on a wider energy scale. Right inset: spin-resolved contributions to the spectral density of the $3d_{3z^2-r^2}$ states. Parameters as in Fig. 2, with the temperature $T = 0.4$ K. See Supplementary Fig. 5 for a direct comparison with experiments.

in the contact regime, the widths of $dI/d(eV)$ and $\rho_1(\omega)$ may differ due to non-equilibrium effects[68]).

**Temperature dependence**. Assuming that the molecule is at equilibrium with the substrate, the observed differential conductance is proportional to

$$\frac{dI(V)}{dV} \propto \int d\omega \rho_h(\omega) f'(\omega - eV), \qquad (7)$$

where $f'(\omega)$ is the derivative of the Fermi function and $\rho_h(\omega)$ is the density of a mixed state which contains the orbitals that have a non-vanishing hopping to the STM tip with an amplitude proportional to the hopping[69]. In the present case we take only two contributions: the $3d_{3z^2-r^2}$ and the conduction states of the same symmetry. Assuming that the latter corresponds to a flat band in the absence of hybridization with the localized states, one has[70]

$$\rho_h(\omega) \propto \sum_\sigma \left[ (1-q^2) \, \mathrm{Im} G_{1\sigma}^d(\omega) + 2q \, \mathrm{Re} \, G_{1\sigma}^d(\omega) \right], \qquad (8)$$

where $q$ is proportional to the amplitude between the STM tip and the conduction electrons of $3z^2 - r^2$ symmetry and is responsible of the observed asymmetry in the line shape. We take $q = 0.4$, a similar value as taken in refs. [15,71], although the results do not show a high sensitivity to $q$.

The temperature dependence of the dip is shown in Fig. 3. The half width of the dip at $T = 0.4$ K, taken at the average between the minimum $dI/dV$ and the relative maximum near $eV = 0.001$ is 3.1 K, near the reported one[18] 2.7 K. With increasing temperature, the low-energy dip in the spectral density decreases and eventually disappears, as expected. The effect is much more pronounced at low temperatures. The same trend is observed in the experiment[19]. The inset shows the results in a scale of energies similar to that of the experiment (Fig. 2 of ref. [19]). See Supplementary Note 4.

**Magnetic field dependence**. The effect of the magnetic field on the $dI/dV$ spectra reported in ref. [19] is most striking. A moderate field of $B = 11$ T ~ 14 K readily transforms the dip into a peak with a transition near 3 to 4 T. This phenomenon is fully reproduced in our calculations, see Fig. 4. For larger fields $B > 20$ T $(4.65 \times 10^{-3})$ a dip is expected to form at the Fermi level as a consequence of the Zeeman splitting between $\rho_{1\uparrow}(\omega)$ and

$\rho_{1\downarrow}(\omega)$. For the comparison with experiment, we have taken the giromagnetic factor $g = 2$, inside the range of uncertainty of previously reported values (see supplemental material of refs. [18,19]). A detailed comparison with the experiment is contained in Supplementary Note 4.

The interpretation of this spectral transformation is highly non trivial. A field-induced topological transition from a NLFL to an OFL would result in an abrupt change from a dip into a peak, clearly at odds with the continuous evolution shown in Fig. 4 and observed in the experiment. To shed more light on this discrepancy, we investigated the Luttinger integrals entering Eq. (4). This study was performed in terms of the equivalent auxiliary Anderson model (see Methods) and interpreted using Eq. (5).

In Fig. 5a we show the evolution of the phase shifts $\delta_{\tau\sigma}$ with increasing field for $D/D_c > 1$ (as in Figs. 2–4). For $B = 0$, the four-phase shifts $\delta_{\tau\sigma} = 0$ (mod. $\pi$) and, therefore, the corresponding four Fermi-level spectral densities $\rho_{\tau\sigma}(0) = 0$ [see Eq. (3)]. As $B$ increases, the phase shifts for the $\pi$ molecular orbital ($\tau = -1$) change only moderately (below $0.12\pi$) and, therefore, the corresponding spectral densities continue to exhibit a dip at the Fermi level. By contrast, the phase shifts that correspond to the $3z^2 - r^2$ molecular orbital ($\tau = 1$) change considerably, reaching the values $\pm\pi/2$ for $B/D_c \sim 0.8$. At this point the spectral densities $\rho_{1\sigma}(0)$ reach their maximum value. Since $\rho_{1\sigma}$ are the main contribution to the differential conductance, the above mechanism explains the transformation from a dip into a peak displayed in Fig. 4. Further increase of $B$ would lead to a Zeeman splitting, as expected for $B \gtrsim T_K^{(1)}$: the peak in $\rho_{1\uparrow}(\omega)$ displaces to lower energies and that in $\rho_{1\downarrow}(\omega)$ to higher energies, so that $\rho_{1\sigma}(0)$ decrease for both $\sigma$, which corresponds to $|\delta_{1\sigma}|$ increasing beyond $\pi/2$ in agreement with Eq. (3).

The changes in the spectral densities at the Fermi level with the magnetic field should lead to important spatial variations of the differential conductance at a small bias. For $B = 0$ in the Kondo limit (integer total occupancy), the spectral densities of the localized orbitals with $\pi$ and $3z^2 - r^2$ symmetry vanish at the Fermi surface and the space variation is dominated by conduction states. As $B$ increases, the influence of $\pi$ orbitals remains small but that of the cylindrical symmetric $3z^2 - r^2$ orbitals increase considerably and therefore, an evolution to a more circular shape is expected, as observed experimentally[19].

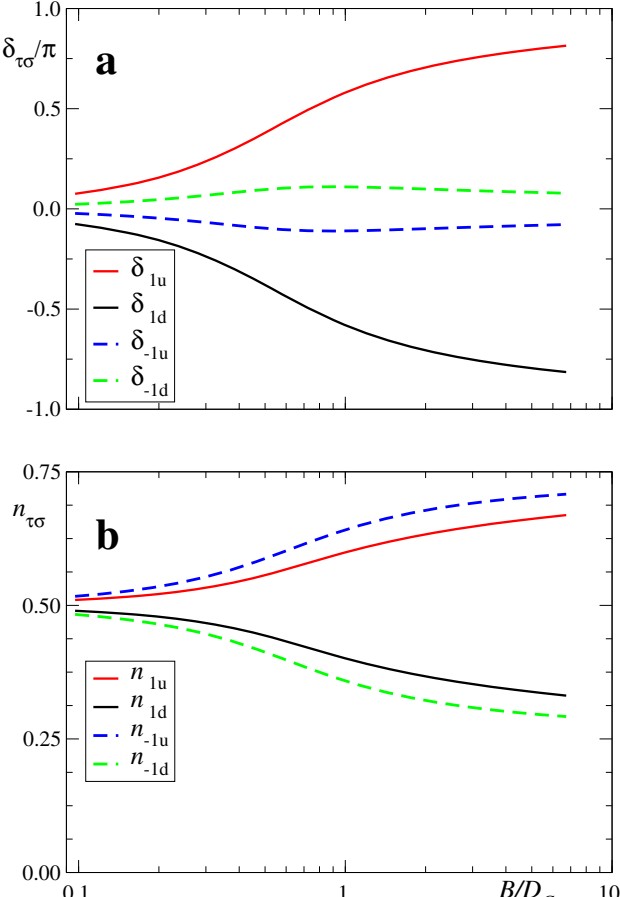

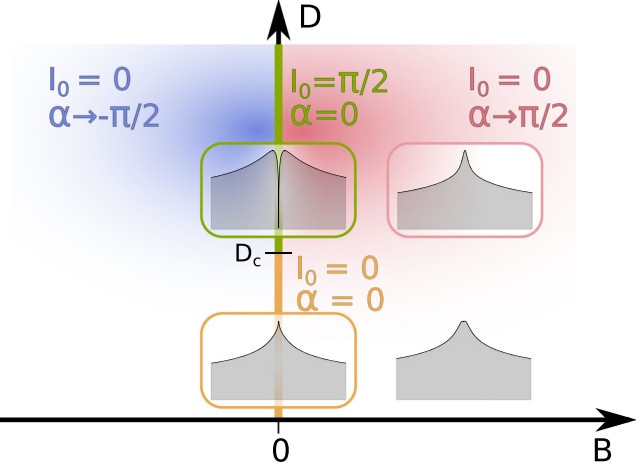

**Fig. 6 Schematic phase diagram of the topological properties.** Phase diagram of the auxiliary Anderson model as a function of the magnetic field $B$ and the magnetic anisotropy $D$. Along the green half line $B = 0$, $D > D_c$, the system is a non-Landau Fermi liquid (NLFL) with the topological value $I_0 = \pi/2$, and the non-topological parameter $\alpha = 0$. For an infinitesimal $B$ both quantities jump as indicated in the figure, yet all physical quantities are continuous. For $D < D_c$, $I_0 = 0$, and $\alpha$ is continuous across $B = 0$, where its value crosses zero (orange segment). The insets show the low-energy region around the Fermi level: regular Fermi liquid (orange frame) and NLFL (green frame) at zero magnetic field, as well as weakly spin-polarized Fermi liquids at moderate magnetic field (white and pink frame).

**Fig. 5 Evolution of impurity scattering and occupancy parameters. a** Phase shifts $\delta_{\tau\sigma}$ and **b** occupancies $n_{\tau\sigma}$ in the auxiliary Anderson model as a function of magnetic field $B$ for $D/D_c = 1.67$. $D_c$ is the anisotropy at the transition point.

The topological properties under an applied magnetic field are very peculiar. For $B = 0$ and $D > D_c$, there is a pole in the self-energy $\Sigma(\omega)$ on the real axis at $\omega = 0$. An infinitesimal $B$ displaces this pole away from $\omega = 0$, as it is shown in Supplementary Note 3. $I_0$ drops to zero and the system becomes an OFL. However, at the same time $\alpha$ jumps to $\pi/2$ ($-\pi/2$) for infinitesimal positive (negative) $B$. See Fig. 6. As a result, in both cases the Luttinger integral $I_{-1\sigma}$ for the majority spin [that is $\uparrow$ ($\downarrow$) for positive (negative) $B$] remains $\pi/2$, while $I_{1\sigma}$ jumps from $\pi/2$ to $-\pi/2$, see Eq. (6). This jump in $\pi$ does not affect any physical properties which are continuous across $B = 0$, as expected and in-line with the "modulo $\pi$ ambiguity" in the definition of the scattering phase shifts[58]. The Luttinger integrals for minority spin are obtained by interchanging the orbital index.

For $D < D_c$, $I_0$ and $\alpha$ are continuous as functions of $B$, and both are equal to zero for $B = 0$. In fact, in the whole $(D, B)$ plane, $I_0 \equiv 0$ and $\alpha$ is continuous except on the half-line $B = 0$, $D > D_c$. This line is, hence, a branch-cut for $\alpha$ (Fig. 6). Furthermore, the point $D = D_c$, $B = 0$ may be considered as a logarithmic singularity for the function $\alpha(D, B)$ viewed as a function in the complex plane with the argument $z = D + iB$. In Fig. 7 we display $\alpha$ as a function of both $D/D_c$ and $B/D_c$. As explained above for $B \to 0^+$, $\alpha(D, B = 0) = (\pi/2)\theta(D - D_c)$, where $\theta(x)$ is the step function. For small non-zero $B$, $\alpha(D, B)$ still resembles this step function. In fact, $\alpha(D, B) \sim -\frac{1}{2}\text{Im}\ln[D_c - D - iB]$. As $|B|$ further increases, $\alpha(D)$ decreases markedly for $D > D_c$. For $|B| \gg D_c$, $\alpha$ is

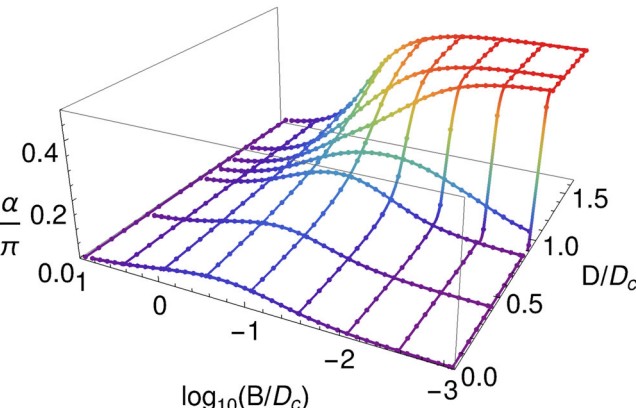

**Fig. 7 Non-topological parameter.** Phase shift $\alpha(D, B)$ for the auxiliary Anderson model as a function of magnetic anisotropy $D$ and magnetic field $B$. The color encodes the value of $\alpha$.

small for all $D$ and as a consequence also the four Luttinger integrals are small.

As $|B|$ increases, the occupancies $\langle n_{\tau\sigma}\rangle$ increase (Fig. 5b), with the effect of increasing $\delta_{\tau\uparrow}$ [Eq. (4)]. For $\delta_{-1\uparrow}$, this effect is largely compensated by the decrease in $\alpha$, leading to a small overall variation of the phase shift. Instead, for $\delta_{1\uparrow}$ both effects add up and $\delta_{1\uparrow}$ has a marked increase, which is then reflected in the transformation of the dip into a peak. For spin $\downarrow$ the effects are opposite.

## Discussion

The ability to flip the differential conductance between low and high values by tuning some external parameter is of tremendous practical importance in molecular electronics for switching device applications. Recently a topological quantum phase transition has been found in a spin-1 KM with two degenerate channels and

single-ion magnetic anisotropy, in which the zero-bias differential conductance and the spectral density at zero temperature jump from their maximum possible values to zero as the longitudinal single-ion anisotropy $D$ is increased beyond a threshold value $D_c$[27,28]. By generalizing the theory of such topological transitions to nonequivalent channels and finite magnetic field, we have for the first time conclusively identified FePc/Au(111) as an experimental realization of this phenomenon, since our approach provides a unified description of the totality of experimental observations. The main difficulty in this identification is that for $D > D_c$ the system is an unconventional Fermi liquid that cannot be adiabatically connected to a non-interacting system by turning off the interactions[27,28,36,37]. The corresponding concept of a NLFL remains largely unfamiliar to most of the physics community. In the NLFL, the Friedel sum rule has to be generalized by introducing a topological quantity that has been previously overseen, even though it leads to a dramatic drop in spectral density and differential conductance for $D > D_c$.

By formulating a general theory of the topological aspects of the Friedel sum rule that incorporates the effects of broken channel and spin symmetries, fully corroborated by numerical calculations for a model Hamiltonian, we reliably established that FePc on Au(111) behaves as a NLFL. With a single set of parameters we explained three key experiments on the differential conductance of this system[15,18,19]. The sole assumption (i.e., that the scanning tunneling microscope senses mainly the 3d orbital of Fe with $3d^2 - r^2$ symmetry) is simple, physically realistic, and consistent with ab initio calculations[63].

The same model also explains the recent experiments for MnPc on Au(111)[38], which exhibit a dependence on magnetic field with qualitative features similar to those in FePc on Au(111). Alternative explanations proposed for these experiments are questionable or contradict well established facts, like the role of Hund rules in defining ground-state multiplets.

While a moderate magnetic field $B$ of the order of 10 T leads to a continuous transition from very small to very high zero-bias conductance, we predict an abrupt transition for $B = 0$ if the tip is pressed against the molecule changing the regime of the system from the tunneling to the contact one, thereby increasing the Kondo temperatures[66–68].

## Methods

The numerical calculations were performed with the NRG Ljubljana[72,73] implementation of the NRG method[65,74] using the separate conservation of the isospin (axial charge) in each channel $\tau$, as well as the conservation of the $z$-component of the total spin, i.e., $SU(2) \times SU(2) \times U(1)$ symmetry. The calculations for the KM were performed with the discretization parameter $\Lambda = 4$ with the broadening parameter $\alpha = 0.8$ for Figs. 3 and 4. For Fig. 2 the values of $\alpha$ used were 0.1 for $f \leq 0.1$, 0.2 for $f = 0.6$ and 0.7, 0.4 for $f = 0.8$ and 0.9 and 0.6 for $f = 1$. The calculations for the Anderson model have been performed using $\Lambda = 3$ with the broadening parameter $\alpha = 0.3$. We kept up to 10000 multiplets (or up to cutoff 10 in energy units) in the truncation, averaging over $N_z = 4$ different discretization meshes. The spectral functions were computed using the complete Fock space algorithm[75], and the resolution for the Anderson model was improved using the "self-energy trick"[76].

For large interaction $U$, the value of $D_c$ calculated with the Anderson model Eq. (2) coincides with that obtained from the corresponding KM. However, for example for $U = 4$, we obtain that the phase shift modulo $\pi$ calculated directly from the NRG spectrum and that obtained the using the generalized Friedel sum rule Eq. (4) deviate by up to $0.1\pi$ for large $B$. This is due to the fact that Eq. (4) were simplified for the case in which the conduction band width (which we have taken as $2W = 2$) is much larger than all other energies involved (the "wide-band limit"), in such a way that the number of conduction electrons per channel is not modified by the addition of the impurity[34]. To avoid this difficulty, we have used $U = 0.4$ in the calculations with the Anderson model. In this case the maximum deviation between both phase shifts is below $0.02\pi$ and the critical anisotropy is reduced by a factor of the order of 6. However, the physical properties are very similar for the same $D/D_c$ and $B/D_c$.

## Data availability

The datasets generated during the current study are available in the Zenodo repository under accession code https://doi.org/10.5281/zenodo.5506654. The data include figure sources and model definition files for the NRG solver.

## Code availability

The NRG calculations presented in this work have been performed with the NRG Ljubljana code. The source code is available from GitHub, https://github.com/rokzitko/nrgljubljana. A snapshot of the specific version used (8f90ac4) has been posted on Zenodo, https://doi.org/10.5281/zenodo.4841076.

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

## Acknowledgements

R.Ž. acknowledges the support of the Slovenian Research Agency (ARRS) under P1-0044. G.G.B. and L.O.M. are supported by PIP2015 No. 364 of CONICET. A.A.A. is supported by PIP 112-201501-00506 of CONICET, and PICT 2017-2726 and PICT 2018-01546 of the ANPCyT, Argentina.

## Author contributions

A.A.A. and L.O.M. conceived the project, R.Ž. and G.G.B. performed the NRG calculations. A.A.A., R.Ž., and L.O.M. wrote the paper.

## Competing interests

The authors declare no competing interests.
