## [Peer Review File · Nature Communications]

REVIEWER COMMENTS

Reviewer #1 (Remarks to the Author):

The manuscript "Iron phthalocyanine on Au(111) as the first experimental realization of a 'non-Landau' Fermi liquid" suggests an interpretation of existing experiments on FePc in terms of an unconventional strongly interacting Fermi liquid state. The authors claim to have identified the first realization of the so-called non-Landau Fermi liquid previously proposed by some of them. A sufficiently strong anisotropic term in the two-body interaction competes with Kondo screening and brings the systems to a topologically distinct phase with vanishing spectral weight and a delta-like divergence in the self-energy at EF.

I find the manuscript interesting and I believe that it contains an important message. However, I am not sure that it matches the criteria for publication in Nature Communications. Specifically, I have two points of criticism, one regarding the novelty of the theoretical proposal and one about the simplifications made to model FePc on Au(111).

Novelty: I find it hard to single out the actual new physics-content of the present manuscript, compared to the previous works published by some of the authors: PRB 98, 195435 (2018) [cited as Ref. 26 with a typo in the page number], PRB 100, 075434 (2019) [cited as 27] and J. Phys.: Condens. Matter 30, 374003 (2018) [cited as Ref. 38].

Ref. 38 deals with the identical system, FePc on Au(111), but in the calculations D has been neglected. Ref. 26 and 27 are motivated by Ni impurities, but the model and the theoretical analysis are very similar to those of the present work, the physical characterization of the Fermi liquid phases as well as the topological nature of the Luttinger integral are proposed, analyzed and discussed.

In the discussion section of the present manuscript the authors write "Prior to the present work, no experimental realization of such a system has been identified.". The present work is however a theory analysis (of a highly simplified model -- see my comments below) which may suggest the presence of NLFL physics in FePc on Au(111). Being a theory-only paper, it obviously does not provide new experimental evidence for this statement. Therefore, a better and more convincing explanation of why these conclusions were not possible just on the basis of Ref. 26, 27 and 38 is needed, in order to justify publication in Nature Communications.

Modeling: FePc on Au(111) is a rather complicated system where several degrees of freedom of the d-shell participate to the collective nature of the Kondo effect in this multi-orbital problem. The effort made by the authors of simplifying the description and identify the relevant physical ingredients is highly appreciable. However, I have some reservations on the following factors that have not been taken into account in writing down the low-energy Hamiltonian:

-From several DFT studies, we know that the hybridization function for the d-shell of a transition-metal atom in the center of Pc molecules is absolutely far from being constant in energy, as instead assumed by the authors. This is particularly the case of the xz/yz doublet: for these two orbitals $\text{Im}_\Delta(E)$ displays even a not-too-broad peak at E_F , resulting actually in the opposite of a flat situation that is usually considered for modeling the Kondo effect. From QMC calculation it is known that this peculiar form of the xz/yz hybridization and its interference with the z2 channel play a role and cannot be ignored. Further, this way of describing Pc molecules on substrates would simplify all effects of the substrate to the local orbital level position, which it is known not to be the only physical effect on which the Kondo effect in these systems depends.

-The individual orbital occupations within the d-shell of these systems is definitely not given by "ideal" integer values. This is extremely relevant for explaining the trend in the Kondo effect (including the characteristic spatial distribution of the Kondo resonance) upon going from Mn, via Fe and Co, up to Cu. My guess is that this model would give very similar results for all of these four molecular systems, despite their strikingly strong experimental differences. For instance Fe- and Co-Pc show no Kondo peak on Ag(001), at odds with Mn-Pc. A natural explanation for this observation is that the filling of the z2 orbital (the one more prone to display a Kondo resonance, even in a mutually coupled five-orbital system) is increasing from about 1.2 in Mn to about 1.7 in Co, therefore rapidly suppressing the tendency towards Kondo in this orbital. My feeling is that the authors would have to assume fine-tuned and ad-hoc behavior of the model parameters (the value of D for example), in order to explain trends like the one in my previous example.

-The Coulomb tensor is highly approximated. The form S_6 contains way less terms than the full Coulomb interaction for a d-shell and the anisotropic term is not justified in a first-principle way. Since it is the crucial parameter for the topological phase transition which is crucial for the interpretation of Fe-Pc on Au(111), its origin and its limitations should be more deeply analyzed.

I would like to see, both in their reply and in the new version of the manuscript a critical discussion of all these points, possibly even an attempt of releasing some of the approximations made in their calculations. Citing previous publications in which the above points have been taken carefully into account (e.g. the several DFT+QMC papers on transition-metal Pc-molecules on metallic substrates

discussing the role of the hybridization, of the filling of the d-shell and of the Coulomb interaction) will be also necessary, though of course not sufficient.

Reviewer #2 (Remarks to the Author):

The authors study theoretically iron phthalocyanine molecules on Au(111), and claim that they realize a non-Landau local Fermi liquid state by analyzing a simple, yet physically motivated model using NRG. In doing so, they focus on the temperature and magnetic field dependent spectral density, which is directly related to the dI/dV characteristics measured in the experiments. In my opinion, they demonstrate clearly that the peculiar variation of the STM current with temperature and magnetic field can be consistently explained using a simple model with a reasonable set of parameters.

Due to recent interest in topological states and the unexpected appearance of topology in a local quantum impurity experiment, I feel that this work deserves attention and possibly to be published in NC.

However, before concluding, I'd like to ask the authors to clarify several aspects of their work.

1. The authors attempt to explain the experimental STM data on iron phthalocyanine molecules on Au(111) (and also on other systems) only qualitatively. However, from the extended discussion in the main text, it seems that most model parameters for Eq. (2) are mostly known. Would it be possible to provide a quantitative comparison to experimental data, at least for certain observables, i.e. by fitting their curves to experimental data? This would make their claims more convincing and substantial.

2. The authors state that there are 3 topological quantities listed in Eq. (5), which allows them to introduce the parameter α in Eq. (6). What is the physical meaning of α ? Is it also a topological invariant? From Fig. 6., it seems to be quantized in certain parameter regime. Is this accidental or protected by topology?

3. I was not able to follow the authors argument in Sec. D. With magnetic field, the dip in the spectral function transforms into a peak in Fig. 4. They say that "The interpretation of this spectral transformation is highly non trivial. A field-induced topological transition from a NLFL to an OFL

would result in an abrupt change from a dip into a peak, clearly at odds with the continuous evolution shown in Fig. 4 and observed in the experiment".

From the ensuing discussion, it is not clear to me why the transition is not abrupt but continuous. Is it related to the peculiar properties of α ?

Reviewer #3 (Remarks to the Author):

The authors present an analytical and computational analysis of the $S=1$ two-channel Kondo model in the presence of symmetry breaking terms, i.e., a magnetic anisotropy DS_z^2 and a Zeeman term BS_z . The study is motivated twofold: First, it is argued that the parameters D and B can drive phase transitions between different kinds of Fermi-liquids, "ordinary" and topological. Here, the latter are called "non-Landau" Fermi liquids. Second, it is claimed that the concept of non-Landau Fermi liquids is essential to correctly interpret three earlier experiments on the Kondo effect in single molecules.

I find the theoretical research presented in the manuscript interesting and relevant. As was reviewed recently in the Rev. Mod. Phys. 92, 035001, single molecules indeed provide a great laboratory for studying exotic Kondo effects. Further, I have no doubt that this research has been conducted with great expertise. Nevertheless, as it stands I cannot recommend the publication of this manuscript.

My difficulty is mainly with the first claim, which in my view is the most important one made in this work: the experimental discovery of a topologically non-trivial Fermi-liquid. It even reflects in the title. For making such a strong claim, it is in my opinion necessary to provide a detailed comparison of experimental data and theoretical predictions. Due to certain universal aspects of the Kondo phenomenon such a comparison often can be made even on the quantitative level. At present, the authors offer to the reader only a few qualitative paragraphs on the second page of the manuscript. Given the claim made in the title, I see a serious discrepancy between what is announced and what is delivered.

We thank the referees for their careful reading of our manuscript. Motivated by their comments we have improved the manuscript as explained in the following.

In the revised version we present the results obtained using a more realistic model that relates the spectral density of the molecular orbitals with the observed differential conductance dI/dV . Specifically, we include the hopping between the tip and the conduction electrons of z^2 symmetry, which are responsible of the asymmetry in the observed line shape. We further include the effects of a broadening due to the Fermi function (Eqs. 7 and 8 of the new version). We also have changed the value of the anisotropy parameter D to make it better correspond to the experimental value and we have changed the smallest exchange interaction parameter to reproduce the observed spectral dip, while the largest exchange is still determined by the experimentally observed largest Kondo temperature.

We have replaced figures 2 to 4 using these improvements. While the topological aspects of the model are essentially unchanged, the numerical results are in much better agreement with the experiment. Now, the overall shapes and energy scales are well reproduced. In particular, all relevant low-energy scales are reproduced: the width of the dip, the temperature scale at which the dip flattens and the magnetic field at which the dip is transformed to a peak. In this way, we remedy the main criticism pointed out by the three referees, that is, the need of a better comparison with experiments in order to support our claim about the existence of a non-Landau Fermi liquid phase.

We have added some important references suggested by the referees concerning modeling aspects of the system, and we have updated Ref. 38 of Guo et al. on Mn phthalocyanine (MnPc), now published in Nat. Comm. We would like to point out again that the interpretation of the experiments reported in this work contradicts basic physical principles. In particular, it assumes that the singlet is below the triplet contradicting the first Hund rule and first-principles calculations. Instead, the experiments are actually fully consistent with our theory.

We note that the manuscript mentioned above is the fourth relevant experiment on transition-metal phthalocyanines that we know (after two Nat. Comm. and one Phys. Rev. Lett. on FePc, and there are relevant experiments in other systems like nickelocene) in which the unfamiliarity with the non-Landau Fermi liquid state and its properties leads to the search of alternative interpretations that are basically wrong and conflicting between them. It seems that every year new relevant experiments of this kind appear.

As Nature Communications includes the records from the referee process in some of its articles, it can be verified that similar criticisms as ours have been made to previous interpretations of measurements in FePc on Au(111). For example, the reviewer 1 of the paper by Yang et al. Nature Comm. 10, 1038 (2019) [The report can be found in this link: https://static-content.springer.com/esm/art%3A10.1038%2Fs41467-019-11587-x/MediaObjects/41467_2019_11587_MOESM2_ESM.pdf] expressed his/her fundamental concerns about the physical consistency of the theoretical interpretation. Specifically, the reviewer has questioned, similar as we did, the

interpretation of the magnetoresistance behavior based in a rearrangement of the d orbitals. On the other hand, on the more recent paper about MnPc on Au(111) by Guo et al. Nature Commun. 12, 1566 (2021), one of the reviewers expressed, same as us, his/her concern about the singlet character of the ground state [The reports can be found in this link: https://static-content.springer.com/esm/art%3A10.1038%2Fs41467-021-21492-x/MediaObjects/41467_2021_21492_MOESM2_ESM.pdf]

Each experimental work presents a separate theoretical interpretation without a proper comparison with the interpretations of the other works and, put all together, there is a clear lack of consensus. We believe that our work presents a unified and rather simple theory of what happens in FePc on Au(111) and this is a strong support for publication of our paper in Nature Communications.

Below we list and explain the changes in the manuscript and respond in detail to the comments of the referees.

LIST OF CHANGES:

- We have changed the title to: "Iron phthalocyanine on Au(111) is a ``non-Landau" Fermi liquid", to take into account the emphasis put by two of the reviewers about the theoretical character of our paper.
- As was asked by the three reviewers, we have considered a more realistic modeling of the FePc/Au(111) system. For this purpose, we have included in the calculations the interference between d orbitals and conduction bands of z² symmetry, as well as the thermal effects on dI/dV. In this way we have obtained a better agreement with experiments, to support our interpretation of the FePc/Au(111) system as a non-Landau Fermi liquid.
- In order to make a better semi-quantitative comparison with experiments, we have tuned the two exchange interactions J^1 and J^{-1} , and we have taken D from the experimental papers.
- As a consequence of the parameter changes, we have redone Figures 2, 3, and 4.
- As suggested by Reviewer 1, we have added several new references about LDA prediction on FePc and theoretical modeling of transition metal phthalocyanines on metallic surfaces.
- We have added a reference to a very recent review on single-molecule electronic transport, mentioned by Reviewer 3.

REVIEWER COMMENTS

Reviewer #1 (Remarks to the Author):

The manuscript "Iron phthalocyanine on Au(111) as the first experimental realization of a 'non-Landau' Fermi liquid" suggests an interpretation of existing experiments on FePc in terms of an unconventional strongly interacting Fermi liquid state. The authors claim to have identified the first realization of the so-called non-Landau Fermi liquid previously proposed by some of them. A sufficiently strong anisotropic term in the two-body interaction competes with Kondo screening and brings the systems to a topologically distinct phase with vanishing spectral weight and a delta-like divergence in the self-energy at EF.

I find the manuscript interesting and I believe that it contains an important message. However, I am not sure that it matches the criteria for publication in Nature Communications. Specifically, I have two points of criticism, one regarding the novelty of the theoretical proposal and one about the simplifications made to model FePc on Au(111).

We thank the reviewer for finding our manuscript interesting and for clearly understanding the message we try to convey.

Novelty: I find it hard to single out the actual new physics-content of the present manuscript, compared to the previous works published by some of the authors: PRB 98, 195435 (2018) [cited as Ref. 26 with a typo in the page number], PRB 100, 075434 (2019) [cited as 27] and J. Phys.: Condens. Matter 30, 374003 (2018) [cited as Ref. 38].

Ref. 38 deals with the identical system, FePc on Au(111), but in the calculations D has been neglected. Ref. 26 and 27 are motivated by Ni impurities, but the model and the theoretical analysis are very similar to those of the present work, the physical characterization of the Fermi liquid phases as well as the topological nature of the Luttinger integral are proposed, analyzed and discussed.

In the discussion section of the present manuscript the authors write "Prior to the present work, no experimental realization of such a system has been identified.". The present work is however a theory analysis (of a highly simplified model -- see my comments below) which may suggest the presence of NLFL physics in FePc on Au(111). Being a theory-only paper, it obviously does not provide new experimental evidence for this statement. Therefore, a better and more convincing explanation of why these conclusions were not possible just on the basis of Ref. 26, 27 and 38 is needed, in order to justify publication in Nature Communications.

In order not to mislead readers, making them think that new experimental measurements are presented in this manuscript, we have changed its title to "Iron phthalocyanine on Au(111) is a "non-Landau" Fermi liquid".

Concerning the previous theoretical studies, it should be noticed that, while former Ref. 38 (39 in the new version) deals with FePc on Au(111), since D was neglected (following misleading arguments of Minamitani et al., Ref. 15 (new

version), who suggested that it could be neglected because it is smaller than TK1), the system is always in the regime of the conventional (non-topological) Fermi liquid. On the other hand, effectively former Refs. 26 and 27 (27 and 28 now) introduce the notions of a topological non-Landau Fermi liquid and that of an anisotropy-driven quantum phase topological transition between two topologically different Fermi liquids, without presenting any existing experimental system that matches the theoretical expectations. However, the theoretical framework proposed in those studies has the following limitations that prevent its direct applicability to FePc:

- a) The two levels considered are equivalent (equal hybridizations with the conduction bands), so that $\text{TK1}=\text{TK2}$.
- b) The magnetic field is always zero.

As it is shown in detail in our present work, the topological properties of a system with TK1 different from TK2 and under a magnetic field are very different from those discussed in former Refs. 26 and 27. In particular, for the Ni system the functions α shown in Fig. 6 are zero. Therefore, the topological properties found for this system are indeed completely new. Eqs. (5) and (6) are entirely new. It is impossible to predict from the existing literature the topological properties of a system with TK1 different from TK2 , even without the magnetic field. In addition, as explained in the text, the transformation of the observed dI/dV from a dip to a peak with applied magnetic field, and the topological properties behind this, are striking and unexpected, even taking into account the results of former Refs. 26 and 27.

It is worth to mention that the new theoretical results presented in this work are an important contribution to the subject of Luttinger theorem, a major theorem in many-body physics since its introduction by J. Luttinger in his seminal paper of 1960 (Ref. 61 new version: Phys. Rev. 119, 1153, 1960). In particular, the vanishing of the so-called Luttinger integral has been taken as one of the hallmarks of a Fermi liquid after Luttinger's perturbative derivation. In this work we have numerically shown that this result does not hold for non-equivalent multi-orbital systems under a finite magnetic field, something that genuinely surprised us since in the literature, to our knowledge, there was no known conventional Fermi liquid with a non-zero Luttinger integral.

Closely related to what is mentioned above, we can say that our work is an important addendum to the fundamental studies of the Friedel sum rule in generalized impurities made by Yoshimori and Zawadowski several years ago (Ref. 35 new version: J. Phys. C 15, 5241 (1982)). In our work we have shown why they could only demonstrate the existence of restricted Friedel sum rules associated with the conservation laws: if the impurity has non-equivalent orbitals under a magnetic field, the usual Friedel sum rule relation between phase shifts and occupation numbers for each separate channel does not hold even in a conventional Fermi liquid phase. We have found that the Luttinger integrals determine three topological invariants, each one associated with a particular conservation law. In our case only one (T) takes a non-zero value in the topologically non-trivial phase. It would be interesting to find other systems

where the other topological invariants (T_σ , T_τ) could take non-zero values.

Beyond the intrinsic importance of these new theoretical results, we are not considering them as the main goal of this manuscript. Our main goal is to show how the non-Landau Fermi liquid scenario allows us to interpret, in a unified and highly satisfactory way, several relevant scattered experimental measurements on FePc, and to point out its possible relevance to other systems.

Modeling: FePc on Au(111) is a rather complicated system where several degrees of freedom of the d-shell participate to the collective nature of the Kondo effect in this multi-orbital problem. The effort made by the authors of simplifying the description and identify the relevant physical ingredients is highly appreciable. However, I have some reservations on the following factors that have not been taken into account in writing down the low-energy Hamiltonian:

We thank the reviewer for appreciating our effort to find a highly simplified model for FePc on Au(111) that nonetheless contain the essential ingredients that allow to interpret several experiments in a unified way.

-From several DFT studies, we know that the hybridization function for the d-shell of a transition-metal atom in the center of Pc molecules is absolutely far from being constant in energy, as instead assumed by the authors. This is particularly the case of the xz/yz doublet: for these two orbitals $\text{Im}_\Delta(E)$ displays even a not-too-broad peak at E_F , resulting actually in the opposite of a flat situation that is usually considered for modeling the Kondo effect. From QMC calculation it is known that this peculiar form of the xz/yz hybridization and its interference with the z² channel play a role and cannot be ignored. Further, this way of describing Pc molecules on substrates would simplify all effects of the substrate to the local orbital level position, which it is known not to be the only physical effect on which the Kondo effect in these systems depends.

As it is known from the universal properties of the Kondo model, the most important fact that determines the Kondo temperature are the effective exchange (two in our case) and the density of conduction states at the Fermi level. Most details of the conduction bands have some small effect on the shape of the Kondo resonance and the dip. However, they do not affect the most important facts, like the EXISTENCE of a peak or a dip and the topological origin of this fact, which has been overseen in three previous Nat. Comm. papers providing unphysical interpretations.

In order to sensibly affect the Kondo spectra that we are discussing, a not-too-broad peak in the density of states of the Au conduction bands at E_F should have a width of the order of the magnetic field responsible of the dip-to-peak transition (or a few times this), that is, of the order of the width of the dip ~ 5 K. LDA (and its generalizations) studies on FePc show that there are no such extremely sharp peaks at the Fermi level (see, for example, Fig. 4 of ref. 53 (new version)). With respect to the DFT calculations, they should not be confused the Au density of states (with smooth variations in a range of width

2W ~ 1 eV around the Fermi level) with the 3d Fe orbital densities (see, for example, Fig. 4 of Ref. 19 new version, where narrow resonances are present).

For the reason above, we are sure that the impurity model with constant density of states is not a questionable approximation in our treatment and that our main conclusions are not modified by the details of the conduction bands.

-The individual orbital occupations within the d-shell of these systems is definitely not given by "ideal" integer values. This is extremely relevant for explaining the trend in the Kondo effect (including the characteristic spatial distribution of the Kondo resonance) upon going from Mn, via Fe and Co, up to Cu. My guess is that this model would give very similar results for all of these four molecular systems, despite their strikingly strong experimental differences. For instance Fe- and Co-Pc show no Kondo peak on Ag(001), at odds with Mn-Pc. A natural explanation for this observation is that the filling of the z^2 orbital (the one more prone to display a Kondo resonance, even in an mutually coupled five-orbital system) is increasing from about 1.2 in Mn to about 1.7 in Co, therefore rapidly suppressing the tendency towards Kondo in this orbital. My feeling is that the authors would have to assume fine-tuned and ad-hoc behavior of the model parameters (the value of D for example), in order to explain trends like the one in my previous example.

One can study the Anderson model discussed in the text and supplemental material out of the half-filling regime and allow for different occupancies. Our goal was to discuss the case of FePc, for which previous first-principles calculations indicate a filling near one for z^2 and π orbitals. To adjust the filling for different systems, one should simply shift the corresponding on-site energies. The most uncertain parameters are the hybridizations, which can vary sensibly among different surfaces, and also the Fano interference modeling. Quite likely assuming reasonable trends along the transition-metal series one can explain with the Anderson model the whole physics. However, this is beyond the scope of the present manuscript. Our main message, that FePc is a non-Landau Fermi liquid, is not affected by these details. To qualitatively describe the behavior of FePc on Au(111) there is no need to fine-tune model parameters, we only need that the system fulfills $D > D_c$, a condition that is fulfilled thanks to the marked different orbital-conduction band hybridization strengths for dz^2 and $d \pi$ orbitals.

Furthermore, regarding the integer filling, LDA calculations of Stepanow shows that the Fe ion when FePc is on Au(111) has an almost integer electronic occupation of 6 (Ref. 51 new version). There is only a very small mixture with the d^7 configuration. Perhaps, this behaviour has to do with the fact that Au(111) is less reactive than others surfaces, like Ag(001), allowing a better modeling of transition metal phthalocyanine molecules on this surface by means of Anderson impurity-like models.

-The Coulomb tensor is highly approximated. The form S6 contains way less terms than the full Coulomb interaction for a d-shell and the anisotropic term is not justified in a first-principle way. Since it is the crucial parameter for the

topological phase transition which is crucial for the interpretation of Fe-Pc on Au(111), its origin and its limitations should be more deeply analyzed.

Effectively, the Coulomb interaction has more terms than the form assumed. A recent work (Ref. 58 in new version) has shown, for another system -Co adatoms on Cu-, that the full consideration of the Coulomb tensor, even beyond the Kanamori model, may be important to correctly capture the physics at an intermediate scale energy ($\sim TK$ or appreciable fraction of TK). However, the scale energy at which we are working is a very low one, for example, the magnetic field responsible for the dip-to-peak transition is ~ 50 times smaller than TK . At this so low scale, the remaining Coulomb terms should be irrelevant for the configurations we have taken. For example, the actual Coulomb interaction contains a term in which a singlet of two electrons in one of the orbitals is displaced to a singlet of the other orbital (it is like a hopping of singlets). However, the singlets lie very high in energy compared with the triplet states.

Concerning the anisotropy interaction D , its physical origin lies in the relativistic spin-orbit coupling (SOC) of the electrons, as happens with all the spin anisotropies. Its value for FePc has been computed in several ab-initio works, and there is an agreement that, although small ($D \sim 5-10$ meV), it is far from being negligible. In fact, in our work we have shown that the presence of the magnetic anisotropy is responsible for the existence of the non-Landau Fermi liquid and the striking magnetic field dependence of the low energy dI/dV spectra. While D is small compared to other "bare" Hamiltonian parameters and so, at first sight, seems to be irrelevant, at low energy the more meaningful comparison is between D and the Kondo temperature. In any case, the value of D we have taken is not only in agreement with previous calculations, but it is also determined experimentally by the experiments of Hiraoka et al. (Ref. 18, new version) for the case in which the molecule is separated from the substrate.

We believe that the central flaw in the previous interpretations of FePc on Au(111) experiments has to do with overlooking the role of D . For example, in Ref. 15 new version, the authors consider that, because TK is larger than D , the anisotropy can be neglected. However, as our NRG calculations have shown, when the hybridization between the Fe dz^2 orbital and the Au bands is much greater than the one corresponding to the d_{xy} orbitals, the critical D_c can be smaller than D , and, consequently, the system is in the NLFL phase, with a sharp dip inside the Kondo resonance. Another example: in Yang et al (Ref. 19 new version), they treat the role of D in an almost classical way, comparing the anisotropy energy with the Zeeman one, to support their "Kondo-orbital selection" driven by the reorientation of the Fe magnetic moment, disregarding the subtle competition between the Kondo screening and magnetic anisotropy effects that gives rise to the NLFL.

We agree with the reviewer that FePc on Au(111) is a rather complicated system; however our main goal is to show that our minimal Kondo model captures its overall physics (even in a semi-quantitative way), giving us confidence that the model includes all the main physical ingredients. We take as fundamental some ingredients of these models that are predicted by several first-principles calculations: the spin 1 of the FePc molecule (see, for example,

Ref.15, 18, and 19), the stronger hybridization of the d_{z^2} orbital with the conduction band than those of the d_{π} orbitals, and the presence of a non-negligible single-ion anisotropy D . The simplified character of the used Kondo and Anderson models allows us to obtain their solution by means of the numerically exact NRG method, which works very well for all temperatures down to 0 K and allow to compute the spectra directly at real frequencies, with a great precision at the low-energy scale of interest. Beyond the fact that, for the reasons given above, the inclusion of all the 3d orbitals and the full Coulomb tensor would not affect our conclusions, it is worth to mention that such a treatment of our problem is out of range of the present state-of-the art numerical methods. At present, the DFT-QMC (see, for example Ref. 58 new version) resolution does not allow the computation in the low-temperature regime (about two orders of magnitude lower than TK1). In addition, it would be necessary to perform some (approximate) analytical continuation to obtain the real frequency spectra, which is very subtle and error-prone in the case of sharp, overlapping spectral features.

I would like to see, both in their reply and in the new version of the manuscript a critical discussion of all these points, possibly even an attempt of releasing some of the approximations made in their calculations. Citing previous publications in which the above points have been taken carefully into account (e.g. the several DFT+QMC papers on transition-metal Pc-molecules on metallic substrates discussing the role of the hybridization, of the filling of the d -shell and of the Coulomb interaction) will be also necessary, though of course not sufficient.

We have added a paragraph in the Section Results A, indicating that our strategy is to solve exactly, at the low energy scale, a minimal model for FePc/Au(111), but that a consideration of the full complexity of the problem may be necessary for a complete and quantitative description. We have added several citations related to the modeling of transition metal phthalocyanine molecules on metallic surfaces to illustrate this point.

Reviewer #2 (Remarks to the Author):

The authors study theoretically iron phthalocyanine molecules on Au(111), and claim that they realize a non-Landau local Fermi liquid state by analyzing a simple, yet physically motivated model using NRG. In doing so, they focus on the temperature and magnetic field dependent spectral density, which is directly related to the dI/dV characteristics measured in the experiments. In my opinion, they demonstrate clearly that the peculiar variation of the STM current with temperature and magnetic field can be consistently explained using a simple model with a reasonable set of parameters.

Due to recent interest in topological states and the unexpected appearance of

topology in a local quantum impurity experiment, I feel that this work deserves attention and possibly to be published in NC.

We thank the reviewer for her/his opinion about the importance of our work.

However, before concluding, I'd like to ask the authors to clarify several aspects of their work.

1. The authors attempt to explain the experimental STM data on iron phthalocyanine molecules on Au(111) (and also on other systems) only qualitatively. However, from the extended discussion in the main text, it seems that most model parameters for Eq. (2) are mostly known. Would it be possible to provide a quantitative comparison to experimental data, at least for certain observables, i.e. by fitting their curves to experimental data? This would make their claims more convincing and substantial.

We have changed the parameters of the figures to include more realistic values of the parameters. In particular, from experiment (Ref. 18 new version) it turns out that $D=5$ meV, while previous calculations (Ref. 15 new version) suggested larger values. We have modified Tk_1/D accordingly and Tk_1 (J_{-1}) to have a dip in agreement with experiment. We have also included the effect of the z^2 conduction band and the Fermi function on the STM results. The resulting new Figs. 2, 3 and 4 show a much better agreement with experiment. There are minor details remaining, probably due to our simplifying assumptions about the model. In any case, we provide a consistent explanation of several experiments published in Nat. Comm. (more on this in the letter to the Editor).

2. The authors state that there are 3 topological quantities listed in Eq. (5), which allows them to introduce the parameter α in Eq. (6). What is the physical meaning of α ? Is it also a topological invariant? From Fig. 6., it seems to be quantized in certain parameter regime. Is this accidental or protected by topology?

α might be defined as the non topological part of the difference between two Luttinger integrals, while I_0 is the topological part (see Eqs. (6)). For magnetic field $B=0$, $\alpha = 0$. However, in the non-Landau phase for $B=0$ ($D > D_c$) where $I_0 = \pi/2$, an infinitesimal B causes a jump in I_0 to zero and simultaneously a jump in α to $\pi/2$ or $-\pi/2$ depending on the sign of B . Instead for $D < D_c$, α is continuous and tends to zero for $B \rightarrow 0$. Therefore for $B \rightarrow 0$ but B different from zero, $I_0 = 0$ always but α jumps at D_c .

It is worth to mention that, under a magnetic field, α is different from zero even for the conventional Fermi liquid phase ($D < D_c$). This is an important theoretical result as, for decades, a zero Luttinger integral has been taken as one of the hallmarks of Fermi liquid phases.

3. I was not able to follow the authors argument in Sec. D. With magnetic field, the dip in the spectral function transforms into a peak in Fig. 4. They say that "The interpretation of this spectral transformation is highly non trivial. A field-

induced topological transition from a NLFL to an OFL would result in an abrupt change from a dip into a peak, clearly at odds with the continuous evolution shown in Fig. 4 and observed in the experiment".

From the ensuing discussion, it is not clear to me why the transition is not abrupt but continuous. Is it related to the peculiar properties of α ?

Yes, in the presence of magnetic field, as it can be inferred from Fig. 5, the evolution of the phase shifts and Luttinger integrals is continuous and determined by the properties of α . This is remarkable as one would think that the jump of the topological invariant T once the magnetic field is turned on (signaling the NFLF - OFL transition) would imply an abrupt change of the spectra (as happens, for example, at $D=D_c$ when $B=0$).

Reviewer #3 (Remarks to the Author):

The authors present an analytical and computational analysis of the $S=1$ two-channel Kondo model in the presence of symmetry breaking terms, i.e., a magnetic anisotropy DS_z^2 and Zeeman term BS_z . The study is motivated twofold: First, it is argued that the parameters D and B can drive phase transitions between different kinds of Fermi-liquids, "ordinary" and topological. Here, the latter are called "non-Landau" Fermi liquids. Second, it is claimed that the concept of non-Landau Fermi liquids is essential to correctly interpret three earlier experiments on the Kondo effect in single molecules.

I find the theoretical research presented in the manuscript interesting and relevant. As was reviewed recently in the Rev. Mod. Phys. 92, 035001, single molecules indeed provide a great laboratory for studying exotic Kondo effects. Further, I have no doubt that this research has been conducted with great expertise. Nevertheless, as it stands I cannot recommend the publication of this manuscript.

We thank the reviewer for finding our manuscript interesting and relevant for the analysis of exotic Kondo effects.

My difficulty is mainly with the first claim, which in my view is the most important one made in this work: the experimental discovery of a topologically non-trivial Fermi-liquid. It even reflects in the title. For making such a strong claim, it is in my opinion necessary to provide a detailed comparison of experimental data and theoretical predictions. Due to certain universal aspects of the Kondo phenomenon such a comparison often can be made even on the quantitative level. At present, the authors offer to the reader only a few qualitative paragraphs on the second page of the manuscript. Given the claim made in the title, I see a serious discrepancy between what is announced and what is delivered.

Taking into account the comments of the referee we have used a more realistic model for the observed differential conductance dI/dV which includes tunneling between the STM tip and the conduction electrons and the effect of temperature. We have changed the value of D according to experimental

observations and changed J_{-1} accordingly to lead to the experimentally observed dip. These changes do not affect the main physics involved, namely the existence of a non-Landau Fermi liquid for $D < D_c$. However, as it can be seen from the new figures 2-4, the agreement with experiment improves considerably.

Given the serious shortcomings of the alternative interpretations of previous relevant work on FePc (Refs. 15, 18, 19 new version) and the related compound MnPc (Ref. 38 new version), we believe that our claim is strongly supported by our new results.

In order not to mislead readers, making them think that new experimental measurements are presented in this work, we have changed its title to "Iron phthalocyanine on Au(111) is a "non-Landau" Fermi liquid".

REVIEWER COMMENTS

Reviewer #1 (Remarks to the Author):

I have read the detailed answer by the authors and am satisfied with it as well as with the changes to manuscript. Therefore, I recommend publication of this work in Nat. Commun.

Reviewer #2 (Remarks to the Author):

The authors have addressed all my comments and question satisfactorily but the one related to quantitative comparison with experiments. While I really appreciate their effort, I still feel that the current qualitative agreement with 3 experiments is somewhat weak and the demonstration of a more quantitative agreement with any of these 3 experiments would have improved the paper a lot.

For example, when quantum impurity models studies within the NRG are compared to experiments, usually a higher level of quantitative agreement can be reached such as in Nature Physics volume 10, 145–150 (2014), though many other papers could be listed here.

I still feel that the presented results are nice and convincing, therefore, I'm tempted to weakly recommend this paper for publication in NC in spite of the above hiatus. Had the authors included a quantitative comparison with some of the experiments, I'd have no hesitation for a strong recommendation.

Reviewer #3 (Remarks to the Author):

After considering the reply of the authors and reading the manuscript again, I am under the following impression. The paper can be published, provided the authors add two elements assisting the presentation and "marketing" of their results.

(i) I believe that the comparison between theory and experiment would be much easier if the authors would reproduce from earlier publications at least some of the experimental data they are discussing in various places of their manuscript. At several occasions they refer to particular features

of earlier experimental traces and it is much easier to follow what the authors are actually alluding to when seeing the data, perhaps even refurbished so as to highlight the authors' point of concern.

(ii) I would like to encourage the authors to add a phase-diagram that illustratively summarizes the different behaviors expected in the different parameter regimes. Also this element will help "marketing" their results.

We thank the reviewers for their careful assessment of our improved manuscript. Motivated by their remaining comments we have refined the manuscript as explained in the following.

Following the comments by Reviewers #2 and #3, in this version we add a direct comparison of our results with the experimental measurements in the form of side-by-side plots. We believe that these show a very good degree of agreement between the numerical results and the actual spectra despite the complexity of the system.

Following the comment by Reviewer #3, we add a phase diagram that illustratively summarizes the different behaviours of the system as a function of the magnetic anisotropy and the magnetic field.

We hope that these two amendments in the presentation of our results fulfil the expectations of Reviewers #2 and #3. Reviewer #1 recommended the publication of the previous version of the manuscript without any reservations.

Below we list and explain the changes in the manuscript and respond in detail to the comments of the referees.

LIST OF CHANGES:

- In the Supplementary Information (SI), we add direct side-by-side comparisons between the calculated spectral functions and the experimental results (supplementary Figs. S3, S4 and S5). In the main text we refer to these new figures at the appropriate points of discussion.
- We add Fig. 6 with a schematic phase diagram, showing the variation of key parameters I_0 and α in the (D,B) plane, with the corresponding spectra as insets.

REVIEWER COMMENTS

Reviewer #1 (Remarks to the Author):

I have read the detailed answer by the authors and am satisfied with it as well as with the changes to manuscript. Therefore, I recommend publication of this work in Nat. Commun.

We thank the reviewer for a positive evaluation of our work.

Reviewer #2 (Remarks to the Author):

The authors have addressed all my comments and question satisfactorily but the one related to quantitative comparison with experiments. While I really appreciate their effort, I still feel that the current qualitative agreement with 3

experiments is somewhat weak and the demonstration of a more quantitative agreement with any of these 3 experiments would have improved the paper a lot.

We thank the reviewer for this suggestion. We have added Figs. S3, S4 and S5 in SI. We have decided for a side-by-side comparison rather than plotting the theoretical curves on-top of the experimental ones, because this approach obviates the need to fit any model parameters, which was not our goal in this work. We find that this comparison shows a very good degree of agreement on a semi-quantitative level (also see below).

For example, when quantum impurity models studies within the NRG are compared to experiments, usually a higher level of quantitative agreement can be reached such as in Nature Physics volume 10, 145–150 (2014), though many other papers could be listed here.

The degree of quantitative agreement depends on what quantity is being studied. In the cited Nature Physics 10, 145 (2014) they focus on the zero-bias conductivity G . Because this quantity is essentially a thermodynamic (equilibrium) ground-state property of the system, there is a very high degree of universality compared to spectral functions, which also depend on the excitations of the system over an extended energy range and, furthermore, need to be compared to experimental transport measurements that typically include non-equilibrium effects that are not included in the NRG calculations.

In fact, the cited Nature Physics paper does show some comparisons between the finite-bias differential conductivity (dI/dV) and spectral functions in Supplementary Information Figs. S11 and S12. Not surprisingly, the degree of agreement is less good compared to that for the zero-bias G . In fact, it is comparable or lower compared to the level of quantitative agreement that we have achieved in the present work.

I still feel that the presented results are nice and convincing, therefore, I'm tempted to weakly recommend this paper for publication in NC in spite of the above hiatus. Had the authors included a quantitative comparison with some of the experiments, I'd have no hesitation for a strong recommendation.

I hope the Reviewer finds the degree of agreement sufficiently good to recommend the paper for publication in NC, especially considering the nature of quantities being compared, as we discussed above.

Reviewer #3 (Remarks to the Author):

After considering the reply of the authors and reading the manuscript again, I am under the following impression. The paper can be published, provided the

authors add two elements assisting the presentation and "marketing" of their results.

(i) I believe that the comparison between theory and experiment would be much easier if the authors would reproduce from earlier publications at least some of the experimental data they are discussing in various places of their manuscript. At several occasions they refer to particular features of earlier experimental traces and it is much easier to follow what the authors are actually alluding to when seeing the data, perhaps even refurbished so as to highlight the authors' point of concern.

We thank the reviewer for this suggestion. We have added Figs. S3, S4 and S5 in SI, as discussed in our reply to Reviewer #2.

(ii) I would like to encourage the authors to add a phase-diagram that illustratively summarizes the different behaviors expected in the different parameter regimes. Also this element will help "marketing" their results.

We thank the reviewer for this suggestion. We have added Fig. 6 with a schematic phase diagram, showing the variation of key parameters I_0 and α in the (D,B) plane, with the corresponding spectra as insets. We believe that this diagram provides a very simple representation of the essential findings of our work and that it will indeed be helpful to market our results.

REVIEWERS' COMMENTS

Reviewer #3 (Remarks to the Author):

I am fully satisfied with the authors response to my suggestions. They have added Fig. 6, which I find very instructive; the comparison of their theory to experiment is very favorable. I believe this is an excellent piece of work that deserves to be published in Nat. Comm..